# Real-time environmental surveillance of SARS-CoV-2 aerosols

Joseph V. Puthussery ®[1], Dishit P. Ghumra ®[1], Kevin R. McBrearty[2], Brookelyn M. Doherty[2], Benjamin J. Sumlin[1], Amirhossein Sarabandi[1], Anushka Garg Mandal[1,3], Nishit J. Shetty[1,9], Woodrow D. Gardiner[2], Jordan P. Magrecki[2], David L. Brody[4,5], Thomas J. Esparza[4], Traci L. Bricker[6], Adrianus C. M. Boon ®[6,7], Carla M. Yuede ®[8] ✉, John R. Cirrito[2] ✉ & Rajan K. Chakrabarty ®[1] ✉

Real-time surveillance of airborne SARS-CoV-2 virus is a technological gap that has eluded the scientific community since the beginning of the COVID-19 pandemic. Offline air sampling techniques for SARS-CoV-2 detection suffer from longer turnaround times and require skilled labor. Here, we present a proof-of-concept pathogen Air Quality (pAQ) monitor for real-time (5 min time resolution) direct detection of SARS-CoV-2 aerosols. The system synergistically integrates a high flow (~1000 lpm) wet cyclone air sampler and a nanobody-based ultrasensitive micro-immunoelectrode biosensor. The wet cyclone showed comparable or better virus sampling performance than commercially available samplers. Laboratory experiments demonstrate a device sensitivity of 77–83% and a limit of detection of 7-35 viral RNA copies/m$^3$ of air. Our pAQ monitor is suited for point-of-need surveillance of SARS-CoV-2 variants in indoor environments and can be adapted for multiplexed detection of other respiratory pathogens of interest. Widespread adoption of such technology could assist public health officials with implementing rapid disease control measures.

The coronavirus disease 2019 (COVID-19) pandemic which began in December 2019 still plagues countries worldwide, with the World Health Organization reporting over 1.7 million new confirmed cases globally during the first week of January 2023[1]. The severe acute respiratory syndrome coronavirus 2 (SARS-CoV-2) coronavirus causes this disease and is spread through respiratory droplets expelled from infected people during coughing, sneezing, breathing, and speaking.

Airborne transmission is recognized as one of the predominant infection pathways[2,3], hence the rapid infectivity rate and virulent nature of the disease. To combat this rapid spread, governments across the globe adopted policies such as mandatory masking in public spaces, quarantining infected individuals, and social distancing to help reduce the risk of airborne transmission. However, such control measures adversely impacted daily life, with consequences such as air

[1]Center for Aerosol Science and Engineering, Department of Energy, Environmental and Chemical Engineering, Washington University in St. Louis, St. Louis, MO 63130, USA. [2]Department of Neurology, Hope Center for Neurological Disease, Knight Alzheimer's Disease Research Center, Washington University, St. Louis, MO 63110, USA. [3]Department of Chemical Engineering, Indian Institute of Technology Bombay, Mumbai 400076, India. [4]National Institute of Neurological Disorders and Stroke, Bethesda, MD, USA. [5]Department of Neurology, Uniformed Services University of the Health Sciences, Bethesda, MD, USA. [6]Department of Medicine, Washington University, St. Louis, MO 63110, USA. [7]Departments Molecular Microbiology, and Pathology and Immunology, Washington University School of Medicine, St. Louis, MO, USA. [8]Department of Psychiatry, Washington University School of Medicine, St. Louis, MO 63110, USA. [9]Present address: Civil and Environmental Engineering, Virginia Tech, Blacksburg, VA 24061, USA. ✉e-mail: yuedec@wustl.edu; cirritoj@wustl.edu; chakrabarty@wustl.edu

travel restrictions, decreased physical activities, restrictions on large social gatherings, and closure of schools and offices. It took many countries almost 2 years to resume normal activities. However, the fear of infection and the periodic rapid resurgence of the disease, for instance, in late December 2022 in China[4], highlights the unpreparedness of even the largest nations in combatting the airborne spread of pathogens. The unavailability of quick and affordable community-level infection detection protocols has been a limiting factor for policymakers in implementing prompt COVID-19 transmission mitigation strategies. A real-time noninvasive surveillance device that can detect SARS-CoV-2 aerosols directly in the air is a potential solution for infection management strategies and the resumption of normal activities.

Offline air sampling techniques are commonly used for virus aerosol detection, where the sample collection and analysis are done in two stages: first, the virus aerosols are collected using standalone bioaerosol samplers, after which the samples are transported to a lab for further analysis. Recent studies used offline air sampling techniques such as condensation growth-based particle into liquid samplers (PILS), wet-wall cyclone-based PILS, and filter sampling, followed by virus detection using reverse transcription-quantitative polymerase chain reaction (RT-qPCR) to detect the presence of SARS-CoV-2 RNA in the air inside hospitals[5–9], shopping centers[10], public transport[10], residential rooms[11], and even outdoor air[12,13]. While these findings underscore the importance of a surveillance method for detecting airborne viruses to control the spread of the infection, these offline methods have long turnaround time (1–24 h), require skilled labor, and do not provide real-time information, which is needed to take swift control measures to manage the airborne spread of the virus.

To our knowledge, there are no commercially available automated real-time airborne SARS-CoV-2 detection devices. This is mainly because of two technology gaps: first is the requirement of an efficient high-flow virus aerosol sampler that can be integrated into a real-time virus detector. Second is the need for a virus detection protocol that is fast, accurate, and sensitive enough to measure the low concentration of viruses typically found in ambient air. Past studies have shown that samplers operating at high flow rates can consolidate aerosols from a large air volume and provide a concentrated sample for biological characterization[8,14,15]. For instance, Ang et al.[8] detected SARS-CoV-2 RNA in 72% of samples collected using a 150 lpm dry air sampler compared to no virus detected in samples collected using the same sampler operated at 50 lpm inside a COVID patient isolation ward. They attributed the higher sample detection to better virus recovery at higher sampling flow rates. Furthermore, recent studies have demonstrated the application of high flowrate PILS for directly collecting pathogen-laden aerosols into a liquid solution and quantifying using real-time virus detectors[16] or offline[8,14,17,18] techniques. While there has been significant progress in developing high-flow PILS devices, very few studies integrate the PILS with real-time sensors for virus detection[16].

Biosensors have recently gained popularity as a promising affordable alternative to RT-qPCR for detecting SARS-CoV-2, as they are low-cost, rapid, sensitive, and highly specific[19,20]. Immunosensors are affinity ligand-based biosensors in which an immunochemical reaction generates various types of signals (optical, electrochemical, thermometric, or microgravimetric) when they bind to a specific target, allowing them to detect the presence of selected pathogens at low concentrations[21,22]. Several studies have successfully demonstrated the application of biosensors for detecting SARS-CoV-2 in nasal swabs[23,24], saliva[20], and exhaled breath condensate samples[25], and achieved similar or better results as compared to RT-qPCR. However, no peer-reviewed studies have utilized biosensors for detecting airborne SARS-CoV-2.

Here, we present a pathogen Air Quality (pAQ) monitor that couples a custom high-flow batch-type wet-wall cyclone PILS with a llama-derived nanobody raised against the SARS-CoV-2 spike-protein covalently attached to a micro-immunoelectrode (MIE) biosensor for near-real-time detection of SARS-CoV-2 in air with 5 min time resolution. The MIE technology was adapted from an electrochemical biosensor used to detect amyloid-β in the setting of Alzheimer's disease[26–28]. Virus-laden aerosols are directly sampled from the air onto a liquid collection medium in the wet cyclone and transferred to the MIE biosensor unit, which detects and reports the presence of virus within 30 s. The wet cyclone performance was compared with other commercially available low-flow PILS. The pAQ monitor performance and sensitivity were validated in the lab using multiple inactivated SARS-CoV-2 virus variants.

## Results and discussion
### Pathogen air quality (pAQ) monitor: design and working principle
The pAQ monitor comprises a batch-type wet-wall glass cyclone (hereafter referred to as wet cyclone) coupled to an MIE detection unit that houses an automated liquid handling unit and MIE biosensor assembly (Fig. 1). The wet cyclone (Supplementary Fig. 1) is connected to a high-flow vacuum pump (Fein Power Tools, PA, USA) to sample air at -1,000 (±10%) lpm. Prior to air sampling, the cyclone is filled with a predefined volume (-15 mL) of phosphate-buffered saline (PBS) solution. The pressure drop rapidly draws in ambient air through a tangential inlet creating a vortex, which produces a rotating film of PBS liquid on the inner wall of the cyclone[29]. Aerosols entering the wet cyclone impact the inner wetted walls and are collected in the liquid media. Aerosols not captured by the wet cyclone exit from the top and are captured by a high-efficiency particulate absorbing (HEPA) filter. Air is sampled for 5 min, after which the concentrated aerosol + PBS solution is transferred to the MIE detection unit.

The detection unit consists of a submerged biosensor attached to a potentiostat, peristaltic pumps to handle the liquid transfer operations, a microcomputer, reagent reservoirs filled with hypochlorous acid (HOCl), PBS, and a 1% bovine serum albumin solution (BSA) diluted in PBS for sensor calibration. The biosensor uses screen-printed carbon electrodes for detecting the presence of virus aerosols based on the MIE technique developed in the Cirrito Laboratory[26,27]. To detect SARS-CoV-2 virions, a nanobody derived in llamas is covalently attached to the electrode surface[30,31]. It detects the oxidation of tyrosine amino acids present in the spike protein of SARS-CoV-2 (see Supplementary Method 2). The SPCEs are pre-treated using PBS, and the surface is pre-blocked in a solution of 1% BSA to avoid binding of non-specific electroactive species. The MIE biosensor is attached to a Potentiostat (PalmSens BV, The Netherlands), and square wave voltammetry (SWV) is performed in blank and sample solutions. The voltage applied is increased stepwise from 0 to 1 V at a frequency of 15 Hz. At -0.65 V, the tyrosine in the SARS-CoV-2 spike (S) protein is oxidized and detected by the MIE as peak oxidation current. The magnitude of the peak oxidation current at -0.65 V indicates the concentration of the virus in each test sample (see Supplementary Method 3).

### Virus aerosol sampling performance in laboratory
The size-dependent particle recovery inside the wet cyclone used in the pAQ monitor was first calculated using CFD simulations (Supplementary Fig. 2). The CFD model results show that the wet cyclone has >95% collection efficiency for particles >1 μm and a cutoff diameter (where the collection efficiency is 50%) of 0.4 μm.

Subsequently, the wet cyclone virus sampling performance was experimentally compared with two commercially available PILS: a BioSampler® (SKC Inc., USA)[32] and a Liquid Spot Sampler ™ (LSS; Aerosol Devices, USA)[33]. The PILS intercomparison experiments were performed by aerosolizing inactivated Washington strain (WA-1) of the SARS-CoV-2 virus inside a well-mixed 21 m³ sealed stainless steel test chamber (Supplementary Fig. 4). The instruments were set up to sample the air inside the chamber simultaneously for 10 min. Note,

**(a)** pAQ monitor schematic

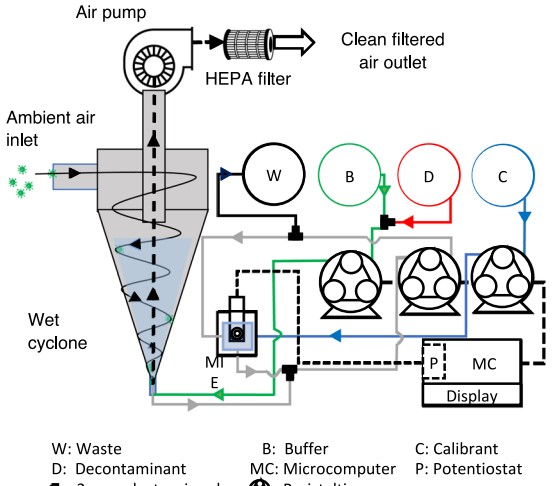

**(b)** pAQ monitor design

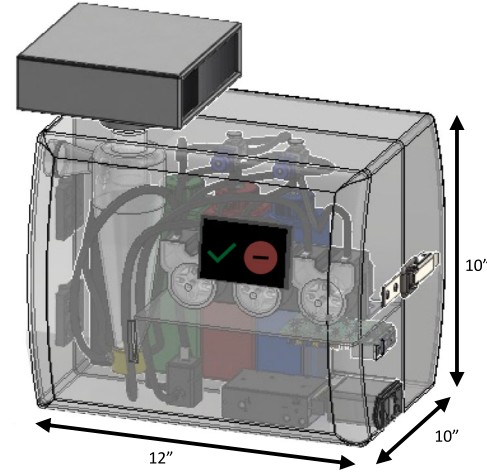

**Fig. 1 | The layout of the pAQ monitor. a** pAQ monitor schematic showing the wet cyclone PILS coupled with the MIE detection unit comprising a submerged MIE biosensor connected to a potentiostat and automated liquid handling accessories, and **b** 3D rendering of the proposed pAQ monitor.

while the high-flow wet cyclone requires <5 min of sampling for virus detection, we performed the chamber experiments for 10 min to ensure sufficient virus concentration was collected in the BioSampler® and LSS for RT-qPCR analysis. After sampling, the virus collected in each device was quantified using RT-qPCR. To study the influence of initial virus concentration on sampling performance, we performed the chamber experiments at three virus loading conditions: <500 copies/m³ ("low"), 500–10,000 copies/m³ ("medium"), and >10,000 copies/m³ ("high"). All experiments were performed either in duplicate or triplicate runs. A detailed description of the experimental setup and protocol is provided in Supplementary Method 5.

Figure 2a compares the virus recovery of the wet cyclone with the BioSampler® and LSS inside a sealed chamber at low, medium, and high aerosolized WA-1 concentrations. After 10 min of sampling, the viral RNA concentration measured by the wet cyclone (i.e., RNA copies/mL of collection media) was, on average, ~10 and ~50 times higher than the concentration measured in the BioSampler® and LSS, respectively. Interestingly, under low concentration conditions, the WA-1 RNA was recovered only in the wet cyclone, whereas the samples collected inside the BioSampler® and LSS were too low to be quantified by RT-qPCR. The high RNA recovery by the wet cyclone can be attributed to its extremely high flow rate, which allows it to sample a larger volume of air (~10 m³) during 10 min sample collection compared to the Bio-Sampler® (~0.125 m³) and LSS (~0.015 m³). This characteristic makes the wet cyclone ideal for use in high-time resolution continuous monitoring applications in real-world environments such as hospitals and patient isolation rooms, where the airborne SARS-CoV-2 RNA concentrations could vary from 2–94,000 copies/m³ (Fig. 2b)[9,34–39]. Under medium and high WA-1 conditions, the air volume normalized RNA concentration measured by the wet cyclone was lower than the concentration reported by the BioSampler®, but was higher than or similar to concentrations determined by the LSS. These results are consistent with the findings of Raynor et al. [14], where they tested the performance of eight air samplers for collecting influenza virus and concluded that high flowrate samplers (>200 lpm) had the highest virus recovery and were ideal for virus detection in an environment with low virus concentrations. However, low flowrate samplers (e.g., BioSampler®) provide a more accurate estimate of the virus concentration in the air. A similar finding was also reported by Luhung et al.[40], where they investigated the effect of increasing the bioaerosol sampler flow rate (100 lpm to 300 lpm) on the bioaerosol recovery and concluded that

high-flow air sampling maximized the time resolution and improved virus capture rate, especially at ultra-low bioaerosol concentrations. However, high-flow sampling is susceptible to inaccurate estimation of bioaerosol concentration per unit air volume. The underestimation of the virus RNA concentration by the wet cyclone in the chamber study could be due to evaporative losses, particle loss to the chamber walls, re-entrainment loss, or particle bounce commonly observed in high-flow wet cyclone sampling[41,42].

## Virus aerosol sampling performance in infected households

We shipped the pAQ monitor assembly to the apartments of two SARS-CoV-2-positive patients for indoor air sampling (Supplementary Method 6). All seven air samples collected using the wet cyclone in the two apartments occupied by SARS-CoV-2 patients tested positive based on RT-qPCR (Fig. 2c). The RT-qPCR results of bedroom samples were compared with air samples collected from a virus-free control room. The Ct values of the air samples from the infected households ranged from 32.7–34.9. In contrast, SARS-CoV-2 RNA was not detected in the control air samples. The significantly lower Ct values observed in the apartment air samples compared to the control air indicate the presence of SARS-CoV-2 RNA in the apartment air. Note, the high Ct value (32.7–34.9) measured suggests that the samples collected were weakly SARS-CoV-2 positive and had very low RNA concentration (see Supplementary Method 8), suggesting low virus aerosol shedding by both volunteers, who self-reported as being asymptomatic during the sampling period. These results are consistent with other studies that have also reported low but statistically significant presence of SARS-CoV-2 in COVID patient isolation rooms and highlight the importance of controlling the air transmission of the virus[5, 11].

## pAQ monitor: limit of detection (LoD) and sensitivity

Figure 3a shows the SARS-CoV-2 variant-specific LoD of the pAQ monitor calculated for 5 min air sampling. The pAQ monitor has an LoD of 35, 7, 9, and 23 RNA copies/m³ of air for the WA-1, delta, beta, and BA-1 strains, respectively. The variability in the LoD is due to the SARS-CoV-2 variant-specific mutations in the spike-protein receptor binding domain epitope that binds to the nanobody, likely varying the nanobody binding efficiency and, thereby, altering the biosensor signal strength. Nevertheless, as evident from Fig. 3a and Supplementary Fig. 3, the signal strength of the MIE biosensor used in the pAQ monitor is sufficient to detect environmentally relevant concentrations of the

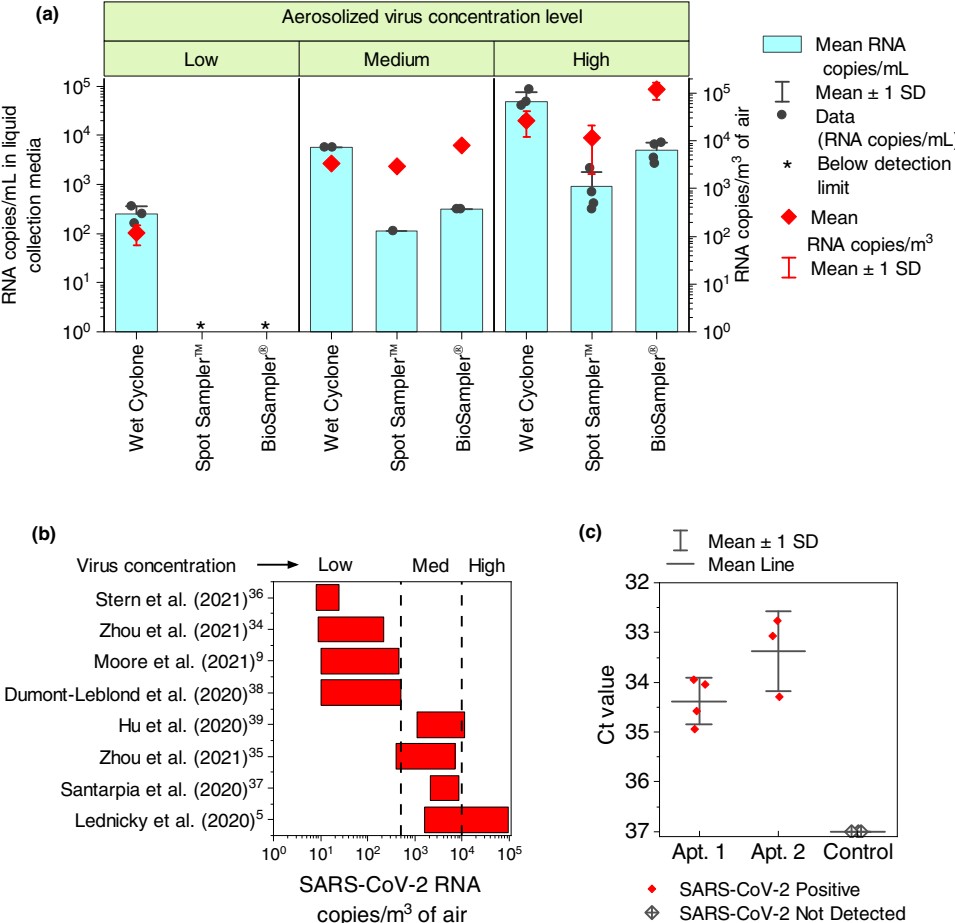

**Fig. 2 | Wet cyclone performance testing. a** Chamber experiments comparing the wet cyclone performance with the BioSampler® and LSS for three aerosolized virus concentration levels: <500 copies/m³ (low; *n* = 3), 500–10,000 copies/m³ (medium; *n* = 2), and >10,000 copies/m³ (high; *n* = 4). The data are presented as mean ± 1 SD of '*n*' independent experiments (**b**) typical concentration of SARS-CoV-2 RNA copies measured in indoor air; the vertical dotted lines demarcate the low, medium and high virus concentration test levels; **c** PCR Ct value (inverted *y*-axis) of indoor air samples collected using the wet cyclone in apartments with SARS-CoV-2-positive patients (*n* = 7) and control room (*n* = 3). The data are presented as mean ± 1 SD of '*n*' independent samples.

four tested SARS-CoV-2 variants, underscoring its use as an environmental SARS-CoV-2 surveillance device. Note that these values only apply for virus aerosols >1 µm (~100% collection efficiency). The LoD for the virus in the submicron-sized aerosols will vary based on the wet cyclone particle size-dependent recovery fraction (Supplementary Fig. 2). Fig. 3b shows the pAQ monitor performance when sampling laboratory aerosolized inactivated WA-1 and BA-1. pAQ monitor showed a sensitivity of 77% for WA-1 and 83.3% for BA-1. The concentrations of WA-1 aerosol samples measured using RT-qPCR are provided in Supplementary Fig. 7. The virus sensitivity of the pAQ is comparable to the sensitivity of other recently developed rapid biosensors (<10 min detection time) used for detecting viruses in saliva[43,44], nasal swabs[45], and exhaled breath condensate[25] samples.

## pAQ monitor: toward real-world deployment

This study demonstrates a proof-of-concept pAQ monitor built by coupling a wet cyclone-based PILS with an ultrasensitive MIE biosensor. The chamber experiments and indoor air sampling inside the apartments of two SARS-CoV-2-positive patients demonstrate the high virus capture efficiency of the wet cyclone even in low virus concentration environments. The high sensitivity (77–83%), high time resolution (5 min), low LoD (7-35 RNA copies/m³), and automation capability of the pAQ monitor make it an ideal choice for affordable (see Supplementary Discussion 1) real-time detection of viruses in different indoor environments such as schools, residences, offices, conference halls, and crowded public places, where real-time virus monitoring (longitudinal

or grab sampling) would enable the occupants to take immediate action to prevent or limit the air transmission of SARS-CoV-2.

A limitation of the proposed pAQ monitor is the high noise level (75–80 dB) during device operation, which can have an adverse effect on the health and comfort of the occupants of a building. Current efforts are underway to find economically feasible solutions to reduce the noise levels to <65 dB, such as using a low-noise motor and soundproofing the device exterior using an acoustic liner. Additionally, we are working on simultaneously detecting other airborne pathogens using the pAQ monitor via multiplexing of MIE biosensors with different target-specific nanobodies. While the findings of this study demonstrate the suitability of the pAQ monitor for real-world applications, the system still requires further testing to verify the robustness of the results in various environments with different aerosol compositions. Future work will focus on comprehensively investigating potential interfering agents in the air that could influence biosensor performance.

## Methods
### pAQ monitor workflow
Before any sample measurement, the biosensor baseline reading is acquired by transferring 2 mL of calibration solution (i.e., 1% BSA in PBS) into the MIE vial and performing square wave voltammetry (SWV). 2 mL of the test aerosol sample is then transferred to the MIE vial, and SWV is performed to measure the oxidation peak height corresponding to the oxidized tyrosine in the viral particle. Tyrosine oxidization occurs at ~0.65 V. The llama-derived nanobody provides specificity against the

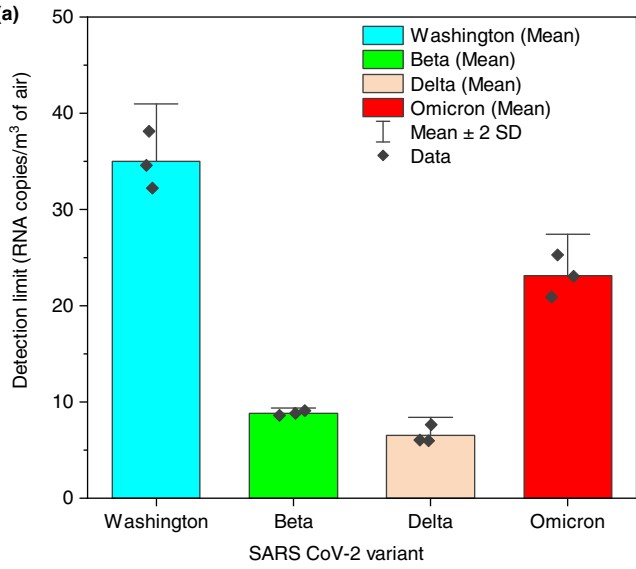

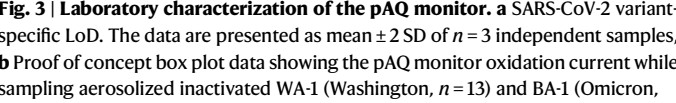

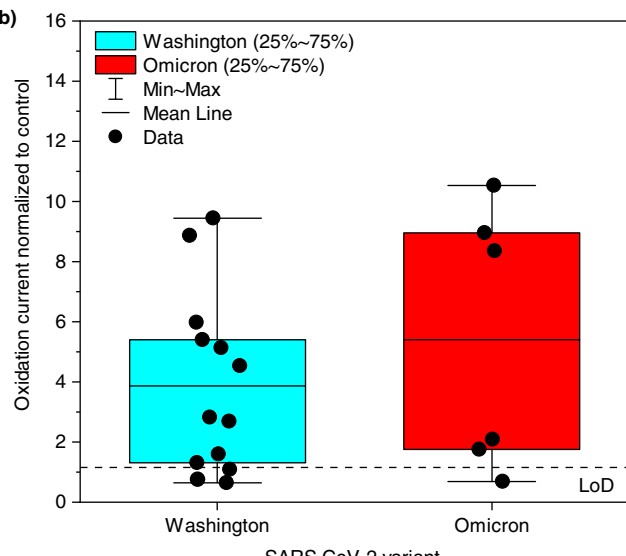

**Fig. 3 | Laboratory characterization of the pAQ monitor. a** SARS-CoV-2 variant-specific LoD. The data are presented as mean ± 2 SD of $n = 3$ independent samples, **b** Proof of concept box plot data showing the pAQ monitor oxidation current while sampling aerosolized inactivated WA-1 (Washington, $n = 13$) and BA-1 (Omicron,

$n = 6$). The box contains 25–75th percentile of the measurements, the center line of the box denotes the mean, and the whiskers denote the minimum and maximum oxidation current measured from '$n$' independent experiments.

SARS-CoV-2 spike-protein. The tyrosine oxidation peak height measured for every aerosol test sample is normalized to the oxidation peak height obtained for virus-free air control to classify the signal as positive or negative reading. After 10 sampling cycles, -15 mL HOCl is injected into the cyclone for decontamination. While the MIE analyzes a sample, the wet cyclone begins collecting the next sample in parallel. Supplementary Methods 2 and 3 provide a description of the biosensor preparation and step-by-step workflow of the pAQ monitor. A 3D rendering of the proposed pAQ monitor made using SolidWorks (Dassault Systèmes, Vélizy-Villacoublay, France) software package is shown in Fig. 1b.

## Computational fluid dynamics to characterize wet cyclone performance

The wet cyclone performance was evaluated numerically using computational fluid dynamics (CFD) software (Ansys Fluent 2021 R1). The size-specific particle tracking inside the cyclone was performed using the Fluent discreet phase method. The fluid flow inside the cyclone was simulated using the Reynold stress model[46]. The assumptions and boundary conditions used in the simulation are provided in Supplementary Method 1.

## Indoor air sampling inside infected households

The wet cyclone assembly (the cyclone, vacuum pump, and PBS solution) was shipped to the apartments of two volunteers who were confirmed SARS-CoV-2 positive. The volunteers collected 5 min air samples ($n = 3$ to 4) from inside their bedrooms/apartments and then stored them in 15 mL centrifuge tubes on ice. The liquid samples were then transported to a laboratory and analyzed using RT-qPCR to detect the presence of SARS-CoV-2. More details on the sampling procedure are provided in Supplementary Method 6.

## pAQ monitor performance evaluation

The Limit of Detection (LoD) of the pAQ monitor is calculated by Eq. (1):

$$\text{LoD}\left(\frac{\text{RNA copies}}{\text{m}^3 \text{ of air}}\right) = \frac{\left[\text{Biosensor LoD}\left(\text{RNA}\frac{\text{copies}}{\text{mL}}\right)\right] * \left[\begin{array}{c}\text{volume of sample inside} \\ \text{the wet cyclone(mL)}\end{array}\right]}{\text{Volume of air sampled(m}^3)} \quad (1)$$

The MIE biosensor LoD was calculated by sequential dilution of a pure stock solution of the inactivated virus. An initial aliquot of the virus of known concentration was diluted sequentially, and the oxidation current ($I_{ox}$) was measured based on SWV. The lowest viral concentrations detected by the biosensor were 32, 8, 6, and 21 RNA copies/mL for the USA/WAa1/2020 (WA1), Beta (B.1.351), Delta (B.1.617.2) and Omicron (BA.1) strains of SARS-CoV-2, respectively (Supplementary Fig. 3). The volume of liquid inside the cyclone after 5-min sampling drops (55-65%) due to evaporation during sampling. For LoD calculation, we took the volume remaining inside the wet cyclone as the average of 10 replicate 5-min sampling experiments.

The pAQ monitor sensitivity calculation, which incorporates the errors from the wet cyclone sample collection and biosensor detection step, was determined by nebulizing inactivated WA-1 and BA-1 (Omicron strain) using a Collison nebulizer and sampling for 5 min using the wet cyclone (Supplementary Method 7). The samples collected were then manually divided into two portions, one was analyzed using the biosensor, and the other was analyzed with RT-qPCR. Due to logistical issues, RT-qPCR was not performed on the BA-1 samples. We used the Origin Pro 2022 software package to perform basic descriptive statistics and generate Fig. 2 and Fig. 3.

## Reverse transcription-quantitative polymerase chain reaction (RT-qPCR)

SARS-CoV-2 viral RNA copies in aerosolized samples were quantified by RT-qPCR based on the method described in Darling et al.[47]. RNA was extracted from 140 μL samples using QIAamp Viral RNA Mini kit (Qiagen) and eluted with 60 μL of Buffer AVE. 8.5 μL RNA was used for real-time RT-qPCR to detect and quantify N gene of SARS-CoV-2 using TaqMan™ RNA-to-CT 1-Step Kit (Thermo Fisher Scientific) on a QuantStudio 12 K Flex Real-time Thermocycler (Applied Biosystems) using the default thermal cycling program. Primers and probes used were 2019-nCoV RUO Kit (IDT). Viral RNA was expressed as N gene copy numbers per mL, based on a standard included in the assay, which was created via in vitro transcription of a synthetic DNA molecule containing the target region of the N gene. Dissociation curves were analyzed following qPCR assay to confirm primer efficacy. Relative mRNA levels were calculated by the comparative Ct method using the ABI 12 K Flex Software package version 1.3. The conversion of Ct

values to the volume of sampled air normalized concentration is provided in Supplementary Method 8.

## Ethics declaration

The Washington University Institutional Review Board deemed this project unnecessary for approval because (1) the study did not collect information about a living human, (2) the study did not involve an interaction or intervention with a living human being performed for research purposes, (3) the study did not involve the collection or use of identifiable, private information, and (4) the study was not testing a device designed to diagnose or treat a medical condition. Informed consent was obtained from the two volunteers to collect and analyze indoor air samples from their apartments.

## Reporting summary

Further information on research design is available in the Nature Portfolio Reporting Summary linked to this article.

## Data availability

The source data are provided as a "Source Data" file. Source data are provided with this paper.

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

## Acknowledgements

This work was funded by the National Institutes of Health (NIH) RADx-Rad program under U01 AA029331 and U01 AA029331-S1 (J.R.C., R.K.C., and C.M.Y.), NIH, National Institute of Neurological Disorders and Stroke (NINDS) Intramural Research Program, the Uniformed Services University of the Health Sciences (D.L.B.); NIH SARS-CoV-2 Assessment of Viral Evolution (SAVE) Program (A.C.M.B.), and WashU-IITB Joint Master's program (J.V.P. and A.G.M.). Y2X Life Sciences has an exclusive option to license this technology for commercialization and were consulted during the design stage. The biosensor used in this study is still in research stage. The corresponding authors can provide the protocol to build and operate the biosensor for interested academic non-commercial groups upon submitting a written request, for non-commercial academic research use. The views expressed in this presentation are those of the authors and do not reflect the official policy or position of the Uniformed Services University, the U.S. Army, the Department of Defense, or the U.S. Government.

## Author contributions

J.R.C., R.K.C., C.M.Y., J.V.P., B.J.S. conceptualized research; J.V.P., D.P.G., K.R.M., W.D.G., B.M.D., J.P.M., A.S., A.G.M. performed research and data analysis. D.L.B., T.J.E., T.L.B., and A.C.M.B. provided expertize and guidance, during experiment design and sample acquisition. J.V.P., D.P.G., R.K.C. wrote the paper with input from J.R.C., C.M.Y., B.J.S., and N.J.S.

## Competing interests

The authors declare no competing interests.
