## [Peer Review File · Nature Communications]

Real-time environmental surveillance of SARS-CoV-2 aerosolsREVIEWER COMMENTS

Reviewer #1 (Remarks to the Author):

The detection of SARS-CoV-2 aerosols is an important issue for environmental surveillance and safety. The device, pathogen air quality monitoring system is thus of interest to test and to obtain more analytical data. However, the development of a nanobody based sensor seems not novel anymore (Communication Medicine 2022 SARS-CoV-2 detection using a nanobody-functionalized voltammetric device) and the paper is largely lacking in depth information about the sensor development and sensor performance

1. Page 11 how come that the control air had also Ct count < 40, which is normally indicative for virus RNA present

2. I lack data to see the correlation of Ct counts and RNA/mL to understand how the RNA copies/m³ were obtained

3. Ct values in Figure 2c are rather high and would be on the limit to be sensed even with the sensor in Comm. Med. As there are no calibration curves for the sensor used it is hard to understand what is possible or not

The device related part of the paper is interesting. The analytical quality of the sensor is lacking information. The same is true for the PCR results.

Reviewer #2 (Remarks to the Author):

The authors describe the development of a proof-of-concept real-time detection system for SARS-CoV-2 aerosols. I found the paper very interesting but must confess to not having sufficient knowledge to adequately review some of the more technical aspects of the study.

The sampling performance of the wet cyclone was compared to two different types of sampler - the BioSampler and Liquid Spot Sampler which, as acknowledged in the text operate at much lower flow rates. Did the authors consider comparing the wet cyclone to a commercially available cyclone sampler e.g. the Coriolis micro (which has been used to detect SARS-CoV-2 in e.g. healthcare environments)?

I found it difficult to fully understand what was being detected at various points of the study and what would be detected if the device was used as intended.

For example, when assessing the cyclone performance in the chamber and inside apartments, the collection fluid was analysed using qPCR detecting RNA not necessarily infectious virus.

Presumably, the biosensor requires an intact virion to oxidise the tyrosine amino acids in the spike protein? Will this detection system only detect intact virions? Will mutations in the spike protein have any impact? Will non-infectious virus be detected? What about 'naked' RNA?

15ml is added to the cyclone which is then operated at 1000 LPM. Are there any losses in volume over the 5-10 min sampling period? Does the liquid remain within the cyclone?

It is stated that 2 ml of the 15ml is analysed and the magnitude of the peak oxidation current indicates the concentration of virus in each (2ml) sample. Is the measurement scalable - can it be used to indicate the concentration of virus in the 5000L air sample?

Chamber studies were carried out using three concentrations of inactivated virus: low, high and medium - the latter ranging from 500-10,000 copies/m³. With such variation how was it determined that replicate experiments were carried out?

Was the temperature and relative humidity within the chamber monitored?

Page 6 (and elsewhere). Would be helpful to define what is meant by "online" and "offline" techniques

Page 9; line 153. I think this should read Figure S2

Page 10; line 168. I think this should read S4

Page 10; line 180. many of the studies cited did not detect infectious virus. Sentence should include "RNA" concentrations

Page 10; line 176. How do the three samplers used compare in terms of sampling efficiency - particularly when detecting particles of the size generated by the Collison?

Page 11; line 194. supplementary info suggests 8 samples were taken (4 in each apartment). 7 measurements illustrated in Figure 2c - what happened to the eighth?

Page 11; line 198. What is considered the cut-off/threshold value for the PCR assay/platform used?

Page 13; line 233. Whilst the wet cyclone detected RNA, it is not known how the sampler performed. No other samplers were used/compared.

Page 13 (Fig 3). I found it difficult to understand this section. Does it relate to the method provided in S7? If so, please clarify in text. Assuming it does - S7 suggests the sample was split and analysed by PCR and using the biosensor. Only data associated with the Washington and Omicron variant is illustrated in 3b. Did the biosensor not detect the Beta and Delta variants? Also the single measurements in 3b seem very variable - does this show a lack of reproducibility?

Page 13; line 237. It is unclear how the authors anticipate this device being used. To obtain snapshot/grab samples or to carry out longitudinal sampling.

Page 15; line 272. S7? The information provided in the supplementary section is essential to understanding the manuscript. Please check it is referred to correctly in the text.

Page 26; lines 561. Risk of residual virus/RNA/contamination impacting subsequent samples?

Page 27; lines 592. How would the taking of control samples work in practice? Presumably sampling would be carried out because aerosolization of SARS-CoV-2 was likely/suspected. How would a virus-free sample be guaranteed?

Page 30. Would be helpful to state the particle size generated by the Collison

Page 33. What was the dominant variant at the time of sampling? What was the vaccination status of the two volunteers?

Reviewer #3 (Remarks to the Author):

The manuscript prepared by Puthussery et al. offers an innovative method for detecting airborne SARS-CoV-2 that combines a wetted-well cyclone air sampler and an MIE biosensor. The authors expect the device to provide close to real-time detection with a 5-minute time resolution. The study has validated the proposed method by comparing it with standard RT-qPCR analysis, and it holds promise for the advancement of biological aerosol detection technologies beyond SARS-CoV-2. However, some aspects of the manuscript could be improved:

It would be helpful to include information on the RNA extraction technique (1-2 sentences) in the main manuscript in addition to the supplement section S8 since RNA extraction is an equally important step to air sampling parameters.

As 1000 lpm is a high flowrate, it would be useful to provide a short discussion on the noise level

generated by the air sampler. The study should also consider whether regular indoor activities can resume in the investigated space, providing a more realistic indoor air sampling scenario.

In line 181-191 and Figure 2a, the difference between "total RNA collected in the liquid" and "RNA normalized by volume of air sampled" is due to the air sampling flowrate. A higher flowrate reduces particle retention efficiency, which affects the purpose of air sampling. If the goal is to estimate the actual concentration of the biological agent in the air, a lower air sampling flowrate should be chosen to provide more accurate estimation. However, if the aim is to improve detection/deal with ultra-low biomass situations, a higher flowrate should be preferred. The authors should refer/cite <https://www.nature.com/articles/s41522-021-00209-4> for this explanation. In Supplementary Figure 2a, the authors found a 20% reduction in particle sampling efficiency after increasing the sampling flowrate from 100 to 300 lpm.

It would be useful to clarify whether the detection limit numbers in Figure 3a are based solely on the known number of inactivated viruses aerosolized. Is there an RT-qPCR result paired with this?

The manuscript states that there is a set of experimental results that compares the two methods (biosensor vs RT-qPCR) from the same sample liquid, but this result is not clearly shown. The authors should clarify whether the biosensor method provides binary results (virus present/absent) or whether it also indicates the quantity of the virus in the sample based on the oxidation current. This relates to comment 4 on whether there is a 1-to-1 quantitation result comparison between the biosensor and RT-qPCR from the same sample liquid.

In line 288, the authors should correct the reference to section S8 as it only explains the extraction kit and RT-qPCR, while section S7 describes the pAQ Monitor Sensitivity Experiments. In Supplementary section S7, it would be useful to provide a reason why the air in the lab is so dry (14-20% RH). A more representative test condition for typical indoor conditions, such as 60-70% RH with 22-25°C, would be preferable.

Lastly, the authors should consider giving insight into the analysis cost per sample of the biosensor relative to the RT-qPCR analysis cost per sample.

REVIEWER COMMENTS

We thank all the reviewers for the detailed and constructive comments which helped significantly improve the manuscript. Our point-by-point responses to all of the comments are listed below. The reviewer comments are in *italics*, the responses are in blue font, and the changes made to the manuscript are shown highlighted using yellow.

Reviewer #1 (Remarks to the Author):

1.1 The detection of SARS-CoV-2 aerosols is an important issue for environmental surveillance and safety. The device, pathogen air quality monitoring system is thus of interest to test and to obtain more analytical data.

Reply. We thank the reviewer for the supportive comments.

1.2 However, the development of a nanobody based sensor seems not novel anymore (Communication Medicine 2022 SARS-CoV-2 detection using a nanobody-functionalized voltammetric device), and the paper is largely lacking in depth information about the sensor development and sensor performance.

Reply. We thank the reviewer for this comment. Here we have summarized the novelty of our study and electrochemical biosensor as well as added a detailed description of the biosensor used in this study.

Since the beginning of the COVID-19 pandemic ~2.5 years back, there has been increased interest in developing reliable, accurate, rapid, and affordable point of care (POC) SARS-CoV-2 detection technologies that could replace commonly used, expensive, and time intense RT-qPCR or the less accurate antigen tests. Biosensor-based virus detection platforms are a promising solution to this technology gap since they are accurate, robust, affordable, and, most importantly, have low turnaround times.

The novelty of this study is the integration of an ultrasensitive nanobody-based biosensor that directly detects the viral particle using a high-flow wet cyclone-based particle into liquid sampler (PILS) that can capture SARS-CoV-2 virus aerosols in 5 min of sampling. As the reviewer pointed out, the study published in “Communication Medicine”,¹ describes a nanobody-functionalized gold electrode sensor for detecting SARS-CoV-2. The sensing system targets the Spike (S) protein of the SARS-CoV-2 virion, similar to our micro-immunoelectrode (MIE) biosensor. However, the target surface receptor and the detection technique are very different. That sensor uses differential pulse voltammetry (DPV) to quantify the signal obtained from VHH-72-13C adapted surface receptor using ferrocene methanol as a redox mediator. The redox mediator serves as an indirect measure of viral particles present. Their sensor has a limit of detection of 12000 copies/mL and typically provides results in 10 minutes of exposure to saliva and nasal swab samples. In contrast, the MIE biosensor demonstrated in our paper uses screen-printed carbon-based electrodes with a nanobody originally derived in llamas covalently bound to the electrode surface to provide specificity to SARS-CoV-2 spike protein. The MIE biosensor uses the SARS-CoV-2 spike protein as the ligand, then uses square wave voltammetry (SWV) to quantify the oxidation of tyrosine

amino acids present SARS-CoV-2 for direct detection of the viral particle. Oxidizing all tyrosines on the exposed SARS-CoV-2 virus also provides endogenous signal amplification, thus increasing the sensitivity of the sensor. The LoD for different SARS-CoV-2 variants varies from 8-32 copies/mL and can provide results in about a 5-minute time resolution. These characteristics differentiate the MIE biosensor from the one described by Pagneux et al. (2022)¹ and other similar sensing systems in the literature.

Furthermore, while nanobody-based sensors have been used in the past for detecting the presence of the virus in saliva or nasal swab samples, to our knowledge, there are no studies integrating a nanobody-based sensor into a real-time environmental air sampler, mainly because of two technology gaps: first is the requirement of an efficient high-flow virus aerosol sampler that can be integrated into a real-time virus detector. Second is the need for a virus detection protocol that is fast, accurate, and sensitive enough to measure the low concentration of viruses typically found in ambient air. We address these two research gaps in this manuscript.

In response to the reviewer's comment, we have added a new section in the supplementary information (SI) section of our revised manuscript with a detailed description of the biosensor used, its specificity and sensitivity in detecting SARS-CoV-2, and the overall working principle.

Pages 5 - 6 in the SI Lines 83 - 120

“Supplementary Method 2: Micro-immunoelectrode (MIE) biosensor

Supplementary Fig. 3 MIE biosensor overview: (a) Working principle, (b) specificity, and (c) sensitivity. The error bar shows the standard error calculated based on five repeat biosensor oxidation current measurements of the same sample.

The MIE biosensor uses screen-printed carbon electrodes (SPiCE, Catalog# SP-1401, BASi Research Products, West Lafayette, IN). A nanobody derived from llamas, which selectively binds to SARS-CoV-2 spike protein, is covalently attached to the electrode surface (Supplementary Fig. 3a). The method for detecting virus aerosols is based on the MIE technique developed in the Cirrito Laboratory.^{2,3} The SARS-CoV-2 detection works based on the electroactivity of Tyrosine amino acids present on the viral particle that can be oxidized on the carbon electrode. The SPiCEs are

pre-treated using phosphate-buffered saline (PBS), and the surface is pre-blocked in a solution of 1% bovine serum albumin (BSA) to avoid binding of non-specific electroactive species. The MIE biosensor uses square wave voltammetry (SWV) to measure the oxidation of tyrosine amino acid present in the viral particle. When SWV is performed, tyrosine oxidation at ~ 0.65 V releases electrons which the sensor detects as current. The height of the peak oxidation current is proportional to the concentration of SARS-CoV-2 virions attached to the electrode surface.

The llama-derived nanobody used in this electrode has a moderate affinity to the SARS-CoV-2 spike (S) protein to allow for dissociation. This ensures that the nanobody binding capacity of the MIE remains open to accept new SARS-CoV-2 spike (S) protein and thereby prolong the longevity of the MIE biosensor. Based on lab characterization, each MIE can be used for ~ 70 sample scans, which equates to ~ 35 real-time sample measurements (i.e., 35 baseline scans+35 sample scans).

We measured biosensor oxidation current at different concentrations of SARS-CoV-1 and SARS-CoV-2 spike proteins. SARS-CoV-1 and SARS-CoV-2 spike proteins have 70% similarity in their genetic makeup. The significantly higher oxidation current measured by the MIE biosensor for SARS-CoV-2 spike proteins (Supplementary Fig. 3b) compared to SARS-CoV-2 spike proteins at concentrations ranging from 0.02 to 2 ng/ml highlights the excellent specificity of the MIE biosensor used in this study.

The MIE biosensor LoD was calculated by sequentially diluting a pure stock solution of the inactivated virus, and the oxidation current (I_{ox}) was measured based on SWV. The lowest viral concentrations detected by the biosensor for the USA/WAa1/2020 (WA1), Beta (B.1.351), Delta (B.1.617.2), and Omicron (BA.1) strains of SARS-CoV-2 are shown in Supplementary Fig. 3c. The LoD for the different strains varied from 6-32 RNA copies/ml. The low LoD obtained shows the ultrasensitive SARS-CoV-2 virus detection capability of the MIE biosensor”

1.3 Page 11 how come that the control air had also Ct count<40, which is normally indicative for virus RNA present. I lack data to see the correlation of Ct counts and RNA/mL to understand how the RNA copies/m³ were obtained Page 11 how come that the control air had also Ct count<40, which is normally indicative for virus RNA present

Reply. To better explain the Ct values presented in this manuscript, we have included the Ct value vs. SARS-CoV-2 RNA concentration calibration curve (Supplementary Figure 8, also shown below) in our revised manuscript. As evident from the calibration curve, Ct values > 37 indicate RNA concentration $\ll 1$ RNA copies/ml, or it can be assumed as the values are very close to the instrument background noise. As pointed out by the reviewer, Ct values > 40 are considered “blank” based on samples that are typically diluted into pure water. In our sampling protocol, the aerosol samples are collected in PBS solution and not pure water, which introduces an inhibitory effect on the PCR steps (Refer to Figure 4 in Zhu et al.⁴). Therefore, our control is not pure water, but instead, PBS was used as the appropriate control and comparison for blank readings. As such, in our assay, **Ct value > 36 is considered background noise**. The average Ct values of the pure PBS control solution for the reported days ($n = 3$; i.e., sample analyses were done on three separate days for three rooms/apartments) of experiments were: Ct= 37.1, 38.4, and 38.9; which were

similar to wet cyclone air control samples taken inside the control room (average Ct = 37.5; Figure 2c in the manuscript).

To avoid confusion while interpreting the results, in our revised manuscript, instead of reporting the average of the “Control” air sample as Ct = 37.5, we report it as “Not Detected.” Based on the calibration curve shown below, a Ct of 37.5 indicates RNA concentration \ll 1 RNA copies/ml and can be assumed to be within the instrument noise levels.

We have added the following discussion in the SI of our revised manuscript, where we explain in detail the step-by-step procedure followed to convert RNA copies from Ct counts to volume-normalized concentrations.

Pages 19 - 20 in the revised SI, Lines 371 - 394

“Supplementary Method 8: RT-qPCR

Supplementary Fig. 8 RT-qPCR Standard Curve. A fresh calibration curve was prepared prior to any RT-qPCR experiments curve to convert the sample Ct values to RNA concentration in copies/ml.

We use the following criteria while reporting the SARS-CoV-2 RNA concentration:

Ct > 36: SARS-CoV-2 not detected.

Ct > 30 and Ct < 36: SARS-CoV-2 positive. These samples can be considered to be weakly positive samples with very low RNA concentrations that cannot be accurately quantified based on our protocol.

Ct <30: SARS-COV-2 positive, and we report the concentration after converting the units to RNA copies/m³ based on the calibration curve shown in Supplementary Fig. 8.

The steps for converting Ct counts to RNA copies/m³ are as follows:

- The Ct value of the virus aerosol samples collected inside the wet cyclone was determined.
- This Ct value was then converted to RNA copies/ml using the calibration curve similar to that shown in Supplementary Fig.8.
- Finally, the concentration in the liquid solution was converted to the volume of air normalized concentration using equation 1 shown below:

$$\text{pAQ Monitor RNA concentration} \left(\frac{\text{RNA copies}}{\text{m}^3 \text{ of air}} \right) = \frac{\left[\text{RNA concentration in collected liquid (copies/ml)} \right]^* \left[\text{volume of liquid inside the wet cyclone (ml)} \right]}{\text{Volume of air sampled (m}^3\text{)}} \quad \dots \text{ eq (1)}$$

The volume of liquid inside the cyclone at the end of sampling was measured manually. The volume of air sampled was calculated by multiplying the sampling flow rate [(~1000 liters per minute (lpm))] with the sampling duration (5 or 10 min).”

1.4 *Ct values in Figure 2c are rather high and would be on the limit to be sensed even with the sensor in Comm. Med. As there are no calibration curves for the sensor used it is hard to understand what is possible or not*

Reply. Thank you for this comment. We would like to clarify that the LoD of the MIE biosensor used in this study is a few orders of magnitude lower than that reported in the Comm. Med. Study. Please see the response to comment 1.2 above (Page 1 of the response document). We agree with the reviewer that the biosensor used in the Comm. Med. Study will not detect the virus in indoor air samples because of its high LoD (~12000 RNA copies/ml).

Note that the entire Fig. 2 panel, including Fig. 2c, shows the ability of the wet cyclone sampler to capture SARS-CoV-2 RNA from ambient air inside bedrooms with infected patients. The high Ct values shown indicate the samples were weakly positive but consistent with several other studies that have reported similar low viral load observed in air samples collected from indoor air (refer to Fig. 2b in the main text).⁵⁻⁸ During the past two years, while several research groups have attempted environmental sampling of SARS-COV-2 from the air, the results have been mixed. For instance, Borges et al.,⁹ reviewed 25 peer-reviewed studies sampling SARS-CoV-2 in an indoor environment (usually in contaminated areas or adjacent high-risk sites), and reported that 12/25 studies did not detect virus aerosols in the air. One of the major reasons for the low success rate in capturing the virus was the choice of aerosol sampler used in the study. The typical concentrations of the virus aerosols (total RNA copies/m³) detected in various studies are shown in Fig. 2b in the main manuscript. Based on these studies, it can be summarized that the detection of SARS-CoV-2 in the air depends on both the ambient level of the virus and the use of a highly efficient bioaerosol sampler. For example, studies collecting viruses on a filter are usually prone to biases associated with extracting the virus from the filter collection media, whereas virus aerosol sampling using high-efficiency samplers such as the SKC BioSampler[®] and Liquid Spot Sampler

(LSS) are constrained by their low sampling flow rates. As a result, these samplers don't perform well when the sampling duration is short or when the room volume is large. High-flow air samplers have shown better success in collecting indoor viruses with low virus aerosol concentrations. An important conclusion from these studies was that the indoor air virus concentration in residential dwellings could be extremely low. This is consistent with our result, where we observed high Ct (>33) values for the 5 min indoor air samples. However, using the wet cyclone, 7/7 air samples tested positive for SARS-CoV-2 based on RT-qPCR. The high virus detection rate by our sampler, even in extremely low virus loading, underscores its applicability to be deployed in various indoor environments with low virus load.

Moreover, in the pAQ monitor system, we couple the wet cyclone with an ultrasensitive MIE biosensor which can detect SARS-CoV-2 as low as 8-21 RNA copies/ml (Supplementary Fig. 3c shown above, page 3 in the response document). This ultrasensitive detection limit of the pAQ monitor makes it an ideal choice for detecting indoor virus aerosols. Note that we did not use the biosensor to measure the virus collected for the data presented in Fig.2c. Therefore, we cannot comment if those virus levels can be detected using our MIE biosensor. Based on the calibration curve shown in Supplementary Fig. 8 in the revised SI, the concentration of virus collected inside the wet cyclone was most likely less than 100 RNA copies/ml. Overall, we can state that if the virus RNA concentration collected inside the wet cyclone is higher than the LoD shown in Supplementary Fig. 3c, the MIE biosensor would be able to detect the presence of the virus accurately.

1.5 The device related part of the paper is interesting. The analytical quality of the sensor is lacking information. The same is true for the PCR results.

Reply. We hope that with the revisions made, and the new information added on biosensor development and RT-qPCR, we have addressed.

References:

1. Pagneux, Q. *et al.* SARS-CoV-2 detection using a nanobody-functionalized voltammetric device. *Commun. Med.* **2**, 1–11 (2022).
2. Yuede, C. M. *et al.* Rapid in vivo measurement of β -amyloid reveals biphasic clearance kinetics in an Alzheimer's mouse model. *J. Exp. Med.* **213**, 677–685 (2016).
3. Prabhulkar, S., Piatyszek, R., Cirrito, J. R., Wu, Z. Z. & Li, C. Z. Microbiosensor for Alzheimer's disease diagnostics: Detection of amyloid beta biomarkers. *J. Neurochem.* **122**, 374–381 (2012).
4. Zhu, Y. *et al.* Printing 2-Dimensional Droplet Array for Single-Cell Reverse Transcription Quantitative PCR Assay with a Microfluidic Robot. *Sci. Rep.* **5**, 1–7 (2015).
5. Zhou, L. *et al.* Breath-, air- and surface-borne SARS-CoV-2 in hospitals. *J. Aerosol Sci.* **152**, 105693 (2021).
6. Stern, R. A., Al-Hemoud, A., Alahmad, B. & Koutrakis, P. Levels and particle size distribution of airborne SARS-CoV-2 at a healthcare facility in Kuwait. *Sci. Total Environ.* **782**, 146799 (2021).
7. Dumont-Leblond, N. *et al.* Low incidence of airborne SARS-CoV-2 in acute care hospital

- rooms with optimized ventilation. *Emerg. Microbes Infect.* **9**, 2597–2605 (2020).
8. Moore, G. *et al.* Detection of SARS-CoV-2 within the healthcare environment: a multi-centre study conducted during the first wave of the COVID-19 outbreak in England. *J. Hosp. Infect.* **108**, 189–196 (2021).
 9. Borges, J. T., Nakada, L. Y. K., Maniero, M. G. & Guimarães, J. R. SARS-CoV-2: a systematic review of indoor air sampling for virus detection. *Environmental Science and Pollution Research* vol. 28 40460–40473 (2021).

Reviewer #2 (Remarks to the Author):

2.1 *The authors describe the development of a proof-of-concept real-time detection system for SARS-CoV-2 aerosols. I found the paper very interesting but must confess to not having sufficient knowledge to adequately review some of the more technical aspects of the study.*

Reply. We thank the reviewer for the helpful comments, which helped us improve our manuscript.

2.2 *The sampling performance of the wet cyclone was compared to two different types of sampler - the BioSampler and Liquid Spot Sampler which, as acknowledged in the text operate at much lower flow rates. Did the authors consider comparing the wet cyclone to a commercially available cyclone sampler e.g. the Coriolis micro (which has been used to detect SARS-CoV-2 in e.g. healthcare environments)?*

Reply. We thank the reviewer for this comment. As we did not have access to a Coriolis micro air sampler, we did not do this performance comparison. However, it would be an interesting future study. Note that the SKC BioSampler® and Liquid Spot Sampler (LSS) used in this study are 2 of the most widely used bioaerosol samplers because of their high bioaerosol collection efficiency. For instance, Eiguren et al.¹ report the particle collection efficiency of the LSS to be greater >95% for particles in the size range of 10 nm to 2500 nm. Similarly, Li et al.² reported a U-shaped physical particle collection of the SKC BioSampler® with >70% recovery for particles larger than 300nm and less than 10 nm in diameter. While we could not find any study directly measuring the size-dependent physical particle collection efficiency of Coriolis micro, Dybwad et al.,³ compared the physical sampling efficiency of the Coriolis micro air sampler with an SKC BioSampler®, and reported the efficiency of the Coriolis micro to be ~58% of that of the reference BioSampler® for 1µm size test particles. We did not find any peer-reviewed literature that reports the size-dependent particle collection of the Coriolis micro in the submicron-sized particle range. However, we expect the collection efficiency of the Coriolis micro air sampler to decrease with a decrease in particle diameter. Considering all these factors, we felt that the SKC BioSampler® and LSS are better reference samplers for this study since we are using a Collison nebulizer that generated aerosols mainly in the submicron range. While we cannot provide a direct comparison of our sampler performance against a Coriolis sampler, we are confident that the overall results will not change since SKC and LSS are excellent reference samplers, as their performance and operation are well characterized and documented in the literature.

We have added the following discussion in the revised manuscript SI:

Page 10, line 224 - 229 of the SI:

“Note that the SKC BioSampler® and LSS used in this experiment are 2 of the most widely used bioaerosol samplers because of their high bioaerosol collection efficiency. For instance, Eiguren et al.¹ report the particle collection efficiency of the LSS to be greater >95% for particles in the size range of 10 nm to 2500 nm. Similarly, Li et al.² reported a U-shaped physical particle

collection of the SKC BioSampler® with >70% recovery for particles larger than 300nm and less than 10 nm in diameter.”

2.3 *I found it difficult to fully understand what was being detected at various points of the study and what would be detected if the device was used as intended. For example, when assessing the cyclone performance in the chamber and inside apartments, the collection fluid was analysed using qPCR detecting RNA not necessarily infectious virus. Presumably, the biosensor requires an intact virion to oxidise the tyrosine amino acids in the spike protein? Will this detection system only detect intact virions? Will mutations in the spike protein have any impact? Will non-infectious virus be detected? What about 'naked' RNA?*

Reply. We thank the reviewer for this comment. To address the comment, we have slightly modified the sentence structuring in the results and discussion section and added more subheadings to help the reader better understand the objective and goal behind each experiment performed. We hope the changes made to the revised manuscript have improved the overall flow of the writing.

Here we would like to clarify that the biosensor used in this study does not differentiate between live and dead viruses. However, the signal is significantly reduced if the virion is not intact. The ultra-sensitivity of the biosensor is not just from oxidation of the spike protein-ligand but from oxidation of all exposed tyrosine on the virion surface. While the MIE biosensor can detect individual spike proteins on fragmented virions, it is with much lower efficiency than in intact virions that contributes tyrosine oxidation from many more proteins in the virion surface than just spike proteins. Because the biosensor requires binding of the spike protein to the virion surface, naked RNA is not detected using this method.

It is correct that the nanobody recognizes a specific epitope on the spike protein that, if mutated, could lose the antigenicity of the nanobody. To date, that epitope has remained largely consistent in all SARS-CoV-2 strains. We have tested inactivated viral variants of SARS-CoV-2 up through BA.1 with minimal change in signal. And in a separate clinical study for breath detection, we have detected the most recent SARS-CoV-2 strains, such as XBB1.5, which is present in St. Louis, MO, area as of April 2023. However, if we encounter a variant with a mutation not recognized by the current nanobody, another nanobody can be identified from our current library and attached to the surface to create a new biosensor. All the lab experiments and pAQ monitor sensitivity analysis was performed using chemically inactivated viruses.

We have added the following discussion in the revised manuscript main text, page 11, lines 210 – 216:

“The variability in the LoD is due to the SARS-CoV-2 variant-specific mutations in the spike protein receptor binding domain epitope that binds to the nanobody, likely varying the nanobody binding efficiency and, thereby, altering the biosensor signal strength. Nevertheless, as evident from Fig. 3a and Supplementary Fig. 3, the signal strength of the MIE biosensor used in the pAQ monitor is sufficient to detect environmentally relevant concentrations of the four tested SARS-CoV-2 variants, underscoring its use as an environmental SARS-CoV-2 surveillance device.”

2.4 15ml is added to the cyclone which is then operated at 1000 LPM. Are there any losses in volume over the 5-10 min sampling period? Does the liquid remain within the cyclone?

Reply. We thank the reviewer for this important comment. Yes, there is significant liquid loss during sampling. In this study, we observed a liquid loss of 55-65% during 5 minutes of air sampling using the wet cyclone. Liquid evaporation loss is a common phenomenon observed in most cyclonic and liquid impingement-type bioaerosol samplers.⁴ The loss in liquid is measured and accounted for when reporting the final virus concentration captured by any PILS (wet cyclone, BioSampler®, and LSS). We accounted for this loss in liquid while calculating pAQ LoD. Note that evaporation during sampling concentrates the virus collected inside the cyclone. However, excess evaporation will lead to insufficient liquid collection media, reducing virus collection efficiency.

We have added the following sentence describing the liquid loss during sampling and how it was accounted for while determining the virus concentration:

Page 16 in the revised manuscript, lines 300 – 302:

“The volume of liquid inside the cyclone after 5-minute sampling drops (55-65%) due to evaporation during sampling. For LoD calculation, we took the volume remaining inside the wet cyclone as the average of 10 replicate 5-minute sampling experiments.”

2.5 It is stated that 2 ml of the 15ml is analysed and the magnitude of the peak oxidation current indicates the concentration of virus in each (2ml) sample. Is the measurement scalable - can it be used to indicate the concentration of virus in the 5000L air sample?

Reply. Yes, all the measurements are scalable, because in wet cyclone sampling, the virus collection in liquid occurs isotropically. The high flow rate and velocity of the air entering the cyclone ensure that the virus aerosols are well-entrained and uniformly mixed in the liquid collection media. This is also evident based on our CFD simulation described in Supplementary Method 1 in the revised manuscript, where we found excellent particle collection inside the cyclone. Therefore, it is safe to assume that a 2 ml aliquot of the collected aerosol liquid sample will have the same concentration as the entire liquid inside the wet cyclone. This concentration can then be normalized by the volume of air sampled to calculate the volume-normalized RNA concentration based on RT-qPCR.

Note that the pAQ monitor is a virus detector, i.e., it reports if the virus is present or absent. While the biosensor oxidation current increase with the increase in the virus concentration, we do not convert this oxidation current to RNA copies per unit volume of air sampled. We assume the sample to be positive if the peak oxidation current after normalizing to control is greater than 1.15 (i.e., > blank + 3SD). However, for the interested readers, we have provided additional results in our revised manuscript (Supplementary Fig. 7), where we show the biosensor oxidation current normalized to control vs. WA-1 RNA concentration measured using RT-qPCR.

2.6 *Chamber studies were carried out using three concentrations of inactivated virus: low, high and medium - the latter ranging from 500-10,000 copies/m³. With such variation how was it determined that replicate experiments were carried out?*

Reply. We thank the reviewer for this question, and we hope to clarify this by explaining the rationale behind choosing the three testing conditions and the main inferences from these experiments.

The three test virus levels assumed in the chamber experiments were determined based on an extensive literature review of recent studies measuring SARS-CoV-2 in indoor air. Our objective was to highlight the wide range of SARS-CoV-2 RNA concentrations in the air reported in these studies. Therefore, to evaluate any environmental sampler performance, it is important that it functions properly at all environmentally relevant concentration levels.

Based on our literature review, we proposed a bioaerosol sampler test classification scheme that could capture this wide range in viral RNA concentration: < 500 copies/m³ (“low”), 500 – 10,000 copies/m³ (“medium”), and >10,000 copies/m³ (“high”). While performing the chamber experiments, we prepared stock virus solutions of known concentrations within these three categories. We tested the three samplers in the “medium” concentration condition twice, and we got similar results in both replicate runs. This is evident from the low standard deviation (<1 of mean %) in the RNA concentration measured in all three devices at the “medium” test conditions. Even though limited, these results show excellent repeatability (standard deviation <1% of mean) of the experiments. However, we observed a larger standard deviation (shown as whiskers in Fig. 2a) for the RNA concentration measured at “High” (n=3) and “Low” (n=4) virus test conditions. We do not have sufficient data to explain the differences in standard deviation between replicate runs for the three test conditions. However, the probable causes include confounding errors associated with chamber wall loss, non-uniform mixing of RNA particles, biases associated with sampler position, air flow rate, sample collection mechanism (i.e., impaction, cyclone, and condensation growth), etc.

Overall, we agree with the reviewer that the number of replicate runs (n=2 to 4) for three test conditions is indeed low. Therefore, we have refrained from making any direct quantitative comparison between the wet cyclone sampling performance and the LSS or BioSampler[®]. However, a qualitative comparison of the sampler performance is valid, based on which we make the inference that a higher RNA fraction is recovered by the wet cyclone (especially at low “virus” test condition) is mainly due to its extremely high flow rate, which allows it to sample a larger volume of air (~ 10 m³) during 10 min sample collection compared to the BioSampler[®] (~ 0.125 m³) and LSS (~ 0.015 m³). These results are consistent with past studies by Raynor et al.,⁵ and Luhung et al.,⁶ as pointed out by Reviewer 2 comment 3.

All the data used in plotting the results (Figures 2 and 3) will be available online with the manuscript.

2.7 *Was the temperature and relative humidity within the chamber monitored?*

Reply. We did not monitor indoor air relative humidity (RH) and temperature. However, the entire chamber is located inside a centrally air-conditioned building (maintained at 70°F), so the initial temperature and RH inside the chamber were the same for each experiment. However, during the aerosolization and sampling process, indoor air conditions might vary, which we did not monitor.

2.8 Page 6 (and elsewhere). Would be helpful to define what is meant by "online" and "offline" techniques

Reply. We thank the reviewer for pointing this out. We define “online” virus measurements as the measurement platforms where air samples are collected and directly transferred to a virus detection unit using some automated liquid delivery system and measured immediately or near-real-time measurements. “Offline” virus aerosol sampling involves first collecting the virus aerosol using a bioaerosol sampler, after which it is removed from the sampler and injected into the virus detector unit. Offline sampling usually is done in batch mode and has longer turnaround times. To simplify the terminologies used, we removed the term “online” from the manuscript and replaced it with real-time virus detector, which is more self-explanatory. The following changes were made in our revised manuscript:

Page 4, lines 66-69

“Offline air sampling techniques are commonly used for virus aerosol detection, where the sample collection and analysis are done in two stages: first, the virus aerosols are collected using standalone bioaerosol samplers, after which the samples are transported to a lab for further analysis. Recent studies used offline air sampling techniques such as condensation growth-based particle into liquid samplers (PILS), wet wall cyclone-based PILS, and filter...”

2.9 Page 9; line 153. I think this should read Figure S2

Reply. We thank the reviewer for pointing this out. We have corrected this in our revised manuscript.

Page 8, line 148

“The size-dependent particle recovery inside the wet cyclone used in the pAQ monitor was first calculated using CFD simulations (Supplementary Fig. 2).”

2.10 Page 10; line 168. I think this should read S4

Reply. We thank the reviewer for pointing this out. We have corrected this in our revised manuscript.

Page 9, lines 164 - 165

“A detailed description of the experimental setup and protocol is provided in Supplementary Method 5.”

2.11 Page 10; line 180. many of the studies cited did not detect infectious virus. Sentence should include "RNA" concentrations

Reply. As per the reviewer's suggestion, we made the following changes to the revised manuscript:

Page 9, lines 177

“...where the airborne SARS-CoV-2 RNA concentrations could vary from...”

2.12 Page 10; line 176. How do the three samplers used compare in terms of sampling efficiency - particularly when detecting particles of the size generated by the Collison?

Reply. In our revised manuscript, we have included the probability distribution function of the aerosol size generated by the Collison nebulizer inside the test chamber. From the size distribution graph, the count median diameter of the aerosols generated was ~50 nm. The LSS and BioSamapler® particle size-dependent efficiency is discussed in our response to comment 2.2 (page 8 of the response document). Briefly, Eiguren et al., 2014¹ report the particle collection efficiency of the LSS to be greater >95% for particles in the size range of 10 nm to 2500 nm. Similarly, Li et al., 2018² reported a U-shaped physical particle collection of the SKC BioSampler® with >70% recovery for particles larger than 300nm and less than 10 nm in diameter.

We added the following Figure to the SI,

Page 13 of the SI, line 281

“

Supplementary Fig. 5 Typical probability density function of aerosol mobility size inside the test chamber during the experiments. This distribution is representative of the SARS-CoV-2 aerosol size distribution found in an indoor environment.”

2.13 Page 11; line 194. *supplementary info suggests 8 samples were taken (4 in each apartment). 7 measurements illustrated in Figure 2c - what happened to the eighth?*

Reply. We thank the reviewer for pointing this out. The discrepancy in the number of samples collected described in the supplementary information was a typo, which we have corrected. Now, the description in the SI matches the method section in the main text and Fig. 2c. A total of 7 air samples were collected from the apartments of the two infected patients.

2.14 Page 11; line 198. What is considered the cut-off/threshold value for the PCR assay/platform used?

Reply. Please refer to our response to reviewer comment 1.3 (page 4 in the response document), where we have provided a detailed description of the RT-qPCR-based RNA quantification used in this study. Briefly, from the RT-qPCR standard curve shown in Supplementary Fig. 8, Ct values > 36 indicate RNA concentration << 1 RNA copies/ml, or it can be assumed as the values are very close to the instrument noise. While interpreting the results, in our revised manuscript, we report samples with Ct > 36 as virus “Not Detected”.

2.15 Page 13; line 233. *Whilst the wet cyclone detected RNA, it is not known how the sampler performed. No other samplers were used/compared.*

Reply. As the reviewer points out, we only used the wet cyclone sampler to collect the 5 min air samples from the households of infected patients. This is mainly due to two reasons:

- (1) **The practical limitation associated with setting up multiple air samplers:** The apartment air sampling experiments inside households of COVID patients were undertaken based on a voluntary basis. When the 2 COVID patients volunteered for this study, the wet cyclone and the associated sampling accessories were immediately shipped to their apartments. Following this, they were trained to operate and collect air samples using the wet cyclone via a video call. While we could have shipped additional bioaerosol samplers to the patient households, we felt it would be rather demanding/tiring for the patient to learn to operate three bioaerosol samplers, considering they were recovering from COVID. Additionally, during the experiments, due to strict social distancing and safety protocols, none of the researchers involved could make direct contact with the patients. Training a volunteer with no prior experience in bioaerosol sampling to operate three instruments over video call was deemed difficult.
- (2) **Bioaerosol Sampler Technology limitations:** The goal of this indoor air sampling experiment was to collect short-duration (5 min) air samples from a real-world indoor environment. As discussed in the manuscript, most of the widely used bioaerosol samplers (e.g., SKC BioSampler® and LSS) operate at low flow rates (< 12.5 lpm), which makes them unsuitable for short-duration air sampling. Even if we had collected air samples using the BioSampler® or LSS, we do not expect to detect any virus because of the low sampling

rate. This is evident from our chamber experiments, where at “low concentration” experiments, the virus RNA was only recovered inside the wet cyclone. The high RNA recovery by the wet cyclone in the chamber experiment can be attributed to its extremely high flow rate, which allows it to sample a larger volume of air (~ 10 m³) during 10 min sample collection compared to the BioSampler® (~ 0.125 m³) and LSS (~ 0.015 m³). This characteristic makes the wet cyclone ideal for high-resolution continuous monitoring applications in real-world indoor environments.

2.16 Page 13 (Fig 3). I found it difficult to understand this section. Does it relate to the method provided in S7? If so, please clarify in text. Assuming it does - S7 suggests the sample was split and analysed by PCR and using the biosensor. Only data associated with the Washington and Omicron variant is illustrated in 3b. Did the biosensor not detect the Beta and Delta variants? Also the single measurements in 3b seem very variable - does this show a lack of reproducibility?

Reply. We thank the reviewer for pointing out this Typo. We have made the following edits to the text in the revised manuscript.

Page 16, lines 303 - 306

“The pAQ monitor sensitivity calculation, which incorporates the errors from the wet cyclone sample collection and biosensor detection step, was determined by nebulizing inactivated WA-1 and BA-1 (Omicron strain) using a Collison nebulizer and sampling for 5 min using the wet cyclone (Supplementary Method 7).”

Due to the limited availability of the inactivated SARS-COV-2 virus strains, we did not have a sufficient quantity of pure stock of inactivated Beta and Delta strains to perform aerosolization experiments to determine the pAQ monitor sensitivity for these two strains. However, we had already characterized the sensitivity of the biosensor against these two stains in the lab by sequential dilution, which allowed us to calculate the pAQ monitor LoD for these two strains (for more details, refer to the response to comment 1.2, Page 2 of this response document).

Here we would like to clarify that, Fig. 3b is not a replicate of a single measurement. The sensitivity analysis plot 3b for WA-1 was obtained by compiling results from 3 different experiments performed on different days, using different MIE electrodes and different aerosolized virus concentrations. This ensures the sensitivity reported incorporates errors associated with MIE biosensor biases, any inter-day measurement/sampling biases, and biases associated with aerosolized virus concentration. We also measured RNA concentration in the wet cyclone collected aerosolized WA-1 samples using RT-qPCR and compared it to the biosensor oxidation current. We found the biosensor signal, in general, increased with an increase in RNA concentration measured using the RT-qPCR (also see our response to Reviewer 3 comment 4, where we addressed a similar comment and Supplementary Figure 7 in the revised SI). Due to the limited availability of the pure stock inactivated BA-1 strain, all the BA-1 aerosolization experiments were completed on the same day using the same electrode but at different aerosolized virus concentrations.

We added the following discussion and Figure in the revised SI:

Pages 18 - 19, lines 349 - 365

“Note that the pAQ monitor sensitivity data for WA-1 was obtained by compiling results from 3 separate experiments performed on different days (number of samples analyzed = 4, 4, and 5 on days 1, 2, and 3, respectively), using different MIE electrodes and different aerosolized virus concentrations. This ensures the sensitivity reported incorporates errors associated with MIE biosensor biases, any inter-day measurement/sampling biases, and biases associated with aerosolized virus concentration. We also measured RNA concentration in the wet cyclone collected aerosolized WA-1 samples using RT-qPCR and compared it to the biosensor oxidation current. biosensor signal, in general, increased with an increase in RNA concentration measured using the RT-qPCR (Supplementary Fig. 7). Due to the limited availability of the pure inactivated BA-1 strain, all the BA-1 aerosolization experiments were completed on the same day using the same electrode but at different aerosolized virus concentrations”.

Figure S7. pAQ Monitor Sensitivity: Biosensor oxidation current normalized vs. WA-1 RNA concentration measured using RT-qPCR. The dashed line indicates the biosensor LoD. Red circles indicate data points marked as outliers and removed prior to the regression analysis.”

2.17 Page 13; line 237. It is unclear how the authors anticipate this device being used. To obtain snapshot/grab samples or to carry out longitudinal sampling.

Reply. As the reviewer mentioned, the pAQ monitor can be used either for grab sampling or for longitudinal sampling. Grab sampling would involve physically moving the pAQ to the desired location and sampling the air in that space. Since the proposed monitor is compact, it can easily be moved from one room to another.

Alternatively, the pAQ monitor can also run continuously at a single location for a fixed time period (i.e., longitudinal sampling). In continuous sampling mode, the device will output 12 virus RNA reading every hour (assuming a time resolution of 5 min). In this mode, the buffer and calibrant bottles will have to be either filled frequently or the reagent bottles will have to be replaced with external large-volume reagent storage containers or reagent bladders.

We added the following text to the revised manuscript:

Page 14, lines 246 – 249

“The high sensitivity (77-83%), high time resolution (5 min), low LoD (7-35 RNA copies/m³), and automation capability of the pAQ monitor make it an ideal choice for affordable (see Supplementary Discussion 1).…”

2.18 Page 15; line 272. S7? The information provided in the supplementary section is essential to understanding the manuscript. Please check it is referred to correctly in the text.

Reply. We thank the reviewer for pointing out this. We have corrected this.

2.19 Page 26; lines 561. Risk of residual virus/RNA/contamination impacting subsequent samples?

Reply. The pAQ monitor has an inbuilt decontamination option as explained in step 7, page 7, line 163 in the SI:

“We have an optional wet cyclone decontamination step where the pump injects 15 ml HOCl into the wet cyclone (~1 minute 30 s) and then rinses it with buffer two times (~ 3 minutes). This can be programmed to happen after every 10 sample collection runs (this frequency can be varied). We expect this periodic decontamination of the wet cyclone would be needed when running the pAQ continuously for several days in an indoor environment with high concentrations of diverse airborne pathogens.”

The frequency of HOCl decontamination, to ensure no residual errors, is user-defined and depends on the sampling mode. For example, if the objective is to collect only a single sample from a room (i.e., grab sampling), we recommend performing a wet cyclone decontamination step before and after every sampling measurement. This will ensure that the data collected is accurate. However, if the pAQ monitor is to run continuously at a single location for 8h a day (i.e., longitudinal sampling), then the decontamination frequency can be reduced. In continuous sampling mode, the device will output 12 virus RNA reading every hour (assuming a time resolution of 5 min). Since the data size is high, the user has the option of averaging multiple data points to get a time-averaged reading. Any residual RNA contamination would not significantly influence the measurements.

For example, if this device is placed inside a large conference hall and operated in continuous mode, the user can define monitor conditions such as, “if three consecutive readings are positive”, then the device can be programmed to sound an alarm, warning the occupants of the potential risk of SARS-CoV-2 virus transmission, and the occupants can be asked to vacate the hall and advised to get tested. Alternatively, the device can be integrated into the building ventilation unit. If three consecutive readings are positive, the device can be programmed to send a signal to the ventilation control unit, which will automatically increase the building ventilation rate to assist in bringing down this virus concentration.

2.20 Page 27; lines 592. *How would the taking of control samples work in practice? Presumably sampling would be carried out because aerosolization of SARS-CoV-2 was likely/suspected. How would a virus-free sample be guaranteed?*

Reply. We thank the reviewer for this question, which is important during the field deployment of the pAQ monitor. There are two options for obtaining control samples:

1. **Pre-calibration:** The MIE biosensors can be pre-calibrated, and baseline preset at the manufacturing facility. Before the pAQ monitor is shipped to the end user, all the biosensors will be calibrated by sampling clean the clean air in a lab or inside a virus-free room to get a baseline reading for the pAQ monitor.
2. **Onsite calibration:** The end user will first do a blank baseline calibration onsite by sampling virus-free air from a well-ventilated room. This is the preferred option since the onsite calibration of the biosensor will ensure the baseline captures any site-specific environmental contaminants that could influence the biosensor oxidation current readings.

2.21 Page 30. *Would be helpful to state the particle size generated by the Collison*

Reply. As per the reviewer's suggestion, we have added the probability distribution of aerosol size in the SI (Supplementary Fig. 5) of the revised manuscript. Please refer to the response to comment 2.12 (page 14 in the response document).

2.22 Page 33. *What was the dominant variant at the time of sampling? What was the vaccination status of the two volunteers?*

Reply. As per the reviewer's comment, we have included this information in the revised supplementary information:

Pages 15 - 16, lines 314-316

“Supplementary Table 3. Summary of the apartment air sampling study”

	Volunteer 1	Volunteer 2
Vaccination status	 • Fully vaccinated 	 • Fully vaccinated • First Dose + Booster shot

	• First Dose + Booster shot	
Dominant SARS-CoV-2 strain ⁷	• BA.5	• BA.5

References

1. Eiguren Fernandez, A., Lewis, G. S. & Hering, S. V. Design and laboratory evaluation of a sequential spot sampler for time-resolved measurement of airborne particle composition. *Aerosol Sci. Technol.* **48**, 655–663 (2014).
2. Li, J. *et al.* Comparing the performance of 3 bioaerosol samplers for influenza virus. *J. Aerosol Sci.* **115**, 133–145 (2018).
3. Dybwad, M., Skogan, G. & Blatny, J. M. Comparative testing and evaluation of nine different air samplers: End-to-end sampling efficiencies as specific performance measurements for bioaerosol applications. *Aerosol Sci. Technol.* **48**, 282–295 (2014).
4. Ghosh, B., Lal, H. & Srivastava, A. Review of bioaerosols in indoor environment with special reference to sampling, analysis and control mechanisms. *Environment International* vol. 85 254–272 (2015).
5. Raynor, P. C. *et al.* Comparison of samplers collecting airborne influenza viruses: 1. Primarily impingers and cyclones. *PLoS One* **16**, e0244977 (2021).
6. Luhung, I. *et al.* Experimental parameters defining ultra-low biomass bioaerosol analysis. *npj Biofilms Microbiomes* **7**, 1–11 (2021).
7. CDC. COVID Data Tracker. *Centers for Disease Control and Prevention* 7–11 https://covid.cdc.gov/covid-data-tracker/#vaccinations-pregnant-women%0Ahttps://covid.cdc.gov/covid-data-tracker/#datatracker-home%0Ahttps://covid.cdc.gov/covid-data-tracker/#cases_casesper100k (2022).

Reviewer #3 (Remarks to the Author):

The manuscript prepared by Puthussery et al. offers an innovative method for detecting airborne SARS-CoV-2 that combines a wetted-well cyclone air sampler and an MIE biosensor. The authors expect the device to provide close to real-time detection with a 5-minute time resolution. The study has validated the proposed method by comparing it with standard RT-qPCR analysis, and it holds promise for the advancement of biological aerosol detection technologies beyond SARS-CoV-2. However, some aspects of the manuscript could be improved:

3.1 It would be helpful to include information on the RNA extraction technique (1-2 sentences) in the main manuscript in addition to the supplement section S8 since RNA extraction is an equally important step to air sampling parameters.

Reply. As per the reviewer's suggestion, we have moved the RNA extraction technique to the methods section of the main manuscript:

Page 16, lines 310 - 319:

“SARS-CoV-2 viral RNA copies in aerosolized samples were quantified by RT-qPCR based on the method described in Darling et al.⁴⁵ RNA was extracted from 140 μ L samples using QIAamp Viral RNA Mini kit (Qiagen) and eluted with 60 μ L of Buffer AVE. 8.5 μ L RNA was used for real-time RT-qPCR to detect and quantify N gene of SARS-CoV-2 using TaqMan™ RNA-to-CT 1-Step Kit (Thermo Fisher Scientific) on a QuantStudio 12K Flex Real-time Thermocycler (Applied Biosystems) using the default thermal cycling program. Primers and probes used were 2019-nCoV RUO Kit (IDT). Viral RNA was expressed as N gene copy numbers per mL, based on a standard included in the assay, which was created via in vitro transcription of a synthetic DNA molecule containing the target region of the N gene. Dissociation curves were analyzed following qPCR assay to confirm primer efficacy. Relative mRNA levels were calculated by the comparative Ct method using the ABI 12K Flex Software package version 1.3. The conversion of Ct values to the volume of sampled air normalized concentration is provided in Supplementary Method 8.”

3.2 As 1000 lpm is a high flowrate, it would be useful to provide a short discussion on the noise level generated by the air sampler. The study should also consider whether regular indoor activities can resume in the investigated space, providing a more realistic indoor air sampling scenario.

Reply. Thanks for this comment. We agree with the reviewer that noise is an important parameter to consider when designing an air quality monitor for indoor air sampling. For the current proof of concept pAQ monitor, we focused on ensuring higher virus detection efficiency over the noise. The main component of the pAQ monitor that generates noise is the motor used for generating the vacuum, which pulls in the air. We tested a couple of high-flow motors to sample air, and the noise

generated ranged from 75 – 80 dB. Note that this device is currently in the proof-of-concept stage and still requires more design modification before it can be used commercially. We are working on finding an engineering solution to reduce noise, such as purchasing a commercial low-noise motor; this will, however, increase the overall cost of our device. We are also exploring the economic feasibility of soundproofing the motor in-house. This is an ongoing task, and our ultimate goal is to bring down the noise to 65 dB (or lower), which is similar to the noise level inside an office.

We have added the following discussion in the revised manuscript:

Page 14, lines 253 – 257

“A limitation of the proposed pAQ monitor is the high noise level (75-80 dB) during device operation, which can have an adverse effect on the health and comfort of the occupants of a building. Current efforts are underway to find economically feasible solutions to reduce the noise levels to <65 dB, such as using a low-noise motor and soundproofing the device exterior using an acoustic liner.”

3.3 *In line 181-191 and Figure 2a, the difference between "total RNA collected in the liquid" and "RNA normalized by volume of air sampled" is due to the air sampling flowrate. A higher flowrate reduces particle retention efficiency, which affects the purpose of air sampling. If the goal is to estimate the actual concentration of the biological agent in the air, a lower air sampling flowrate should be chosen to provide more accurate estimation. However, if the aim is to improve detection/deal with ultra-low biomass situations, a higher flowrate should be preferred. The authors should refer/cite <https://www.nature.com/articles/s41522-021-00209-4> for this explanation. In Supplementary Figure 2a, the authors found a 20% reduction in particle sampling efficiency after increasing the sampling flowrate from 100 to 300 lpm.*

Reply. We thank the reviewer for sharing this study. We have incorporated this interpretation/inference of our results in our revised manuscript:

Page 10, lines 185 - 189:

“. A similar finding was also reported by Luhung et al.,³ where they investigated the effect of increasing the bioaerosol sampler flow rate (100 lpm to 300 lpm) on the bioaerosol recovery and concluded that high-flow air sampling maximized the time resolution and improved virus capture rate, especially at ultra-low bioaerosol concentrations. However, high-flow sampling is susceptible to inaccurate estimation of bioaerosol concentration per unit air volume.”

3.4 *It would be useful to clarify whether the detection limit numbers in Figure 3a are based solely on the known number of inactivated viruses aerosolized. Is there an RT-qPCR result paired with this?*

Reply. We have performed the RT-qPCR for all the data points reported in Figure 3a. Here we would like to clarify, Fig. 3a was obtained using the equation shown below:

$$\text{LoD} \left(\frac{\text{RNA copies}}{\text{m}^3 \text{ of air}} \right) = \frac{\left[\text{Biosensor LoD} \left(\frac{\text{RNA copies}}{\text{ml}} \right) \right] * \left[\text{volume of sample inside the wet cyclone (ml)} \right]}{\text{Volume of air sampled (m}^3\text{)}} \quad \dots (1)$$

The limit of detection (LoD) of the MIE biosensor was calculated by sequential dilution of a pure stock solution of the inactivated virus (Supplementary Fig.3 in the SI of the revised manuscript). An initial aliquot of the virus of known concentration was diluted sequentially, and the oxidation current was measured based on square wave voltammetry. The lowest viral concentrations detected by the biosensor (i.e., the LoD) are shown in Supplementary Fig.3. Additionally, the average sample volume inside the cyclone after every 5 minutes of air sampling was measured multiple times (n=10 independent experiments), and the volume of air sampled was calculated as 1000 lpm ($\pm 10\%$; air flow rate) multiplied by time (i.e., 5 minutes). This allowed us to calculate the LoD of the pAQ monitor, which is a constant and does not depend on the aerosolization technique adopted. Please see our response to reviewer comment 2.16 on pages 16-17 of this response document, where we show the paired RT-qPCR results vs. biosensor signal.

We have added a detailed discussion of the MIE biosensor operation and its LoD calculations based on RT-qPCR in Supplementary Method 2 in the SI of our revised manuscript.

3.5 The manuscript states that there is a set of experimental results that compares the two methods (biosensor vs RT-qPCR) from the same sample liquid, but this result is not clearly shown. The authors should clarify whether the biosensor method provides binary results (virus present/absent) or whether it also indicates the quantity of the virus in the sample based on the oxidation current. This relates to comment 4 on whether there is a 1-to-1 quantitation result comparison between the biosensor and RT-qPCR from the same sample liquid.

Reply. We thank the reviewer for this comment. Please refer to the previous responses (response to comment 2.16 and comment 3.4), where we explain the 1 to 1 comparison of the biosensor and RT-qPCR results. Briefly, the pAQ monitor is designed to provide a binary output; that is, it reports if the virus is present or absent. We also measured the RNA concentration in the wet cyclone collected aerosolized WA-1 samples using RT-qPCR and compared it to the biosensor oxidation current. The biosensor oxidation signal, in general, increased with an increase in RNA concentration measured using the RT-qPCR (Supplementary Fig. 7 in the revised manuscript, also shown on page 17 in this response document). However, the current version of the pAQ Monitor does not quantify the exact virus RNA concentration.

This method to classify the sample as SARS-CoV-2 positive or negative is described on Page 14, line 271-273 of the revised manuscript:

“The tyrosine oxidation peak height measured for every aerosol test sample is normalized to the oxidation peak height obtained for virus-free air control to classify the signal as positive or negative reading.”

The signal classification as positive or negative is also described on Page 7, “Step 3”, lines 152-154 of the revised SI:

“This baseline subtracted peak oxidation current is normalized with a virus-free control oxidation current (explained later). We assume the sample to be positive if the peak oxidation current after normalizing to control is greater than 1.15 (i.e., > blank + 3SD).”

3.6 *In line 288, the authors should correct the reference to section S8 as it only explains the extraction kit and RT-qPCR, while section S7 describes the pAQ Monitor Sensitivity Experiments. In Supplementary section S7, it would be useful to provide a reason why the air in the lab is so dry (14- 20% RH). A more representative test condition for typical indoor conditions, such as 60-70% RH with 22-25°C, would be preferable.*

Reply. We thank the reviewer for pointing this out, and we have corrected the text in the revised manuscript.

Page 16, line 306

“...the wet cyclone (Supplementary Method 7).”

The indoor RH observed in this study is very common in the Midwest USA, as the air can get very dry during the winter months. Such low RH makes the experiment conditions more challenging. This is because the low RH leads to rapid evaporation⁴ of the aerosols generated by the Collison nebulizer and shrinks the aerosol (i.e., the diameter decreases) before it enters the wet cyclone. As shown in Supplementary Fig. 2 in the revised SI, the particle retention efficiency of the wet cyclone is dependent on the aerosol diameter. The lower the particle diameter, the lower would be the collection efficiency. The test conditions assumed in this study can be considered an extreme or worst-case scenario. The sensitivity reported in this manuscript is a conservative estimate; in a more typical RH (60-70%), we can expect a higher virus detection by the pAQ monitor. The particle collection and device sensitivity reported in this study can be assumed as conservative estimates assuming extreme conditions; the actual instrument performance would be much better.

As requested by the reviewer, we have added the following text to our revised SI:

Page 17, line 335 - 340

“Average lab temperature = 70 °F and indoor RH = 14% – 20%. The low RH conditions in our lab could lead to rapid evaporation⁸ of the aerosols generated by the Collison nebulizer and shrink the aerosol before it enters the wet cyclone. This, in turn, would lower the overall particle collection efficiency of the wet cyclone, as shown in Supplementary Fig.2. The device sensitivity reported in this study can be assumed as conservative estimates assuming extreme RH conditions.”

3.7 *Lastly, the authors should consider giving insight into the analysis cost per sample of the biosensor relative to the RT-qPCR analysis cost per sample*

Reply: We thank the reviewer for this question. A preliminary cost analysis is described below. Cost per sample is an important parameter for end users of the pAQ monitors. Here, note that the pAQ monitor is currently in the proof-of-concept stage. The cost analysis is primarily based on

raw materials and design costs for building a single unit of the pAQ monitor. The cost will significantly reduce when the production is scaled up.

Below we provide the cost of building a pAQ monitor in a research laboratory setting which we have also included in the revised SI Page: 21, lines 429 - 468

“Supplementary Discussion 1: pAQ Monitor Cost Analysis

Fixed cost for building a single pAQ Monitor:

- The cost of building a single wet cyclone air sampler, which includes the cost of tubing connectors, labor, design, and shipping = \$600 - \$700
- The cost of building the MIE measurement unit that comprises the biosensor, automated liquid handling pumps, microcontroller, potentiostat measurement setup, and miscellaneous accessories = \$700 - \$1000
- Cost of the air blower/vacuum air pump = \$100 - \$200
- The total cost to build and design a single pAQ unit in a research laboratory will range from \$1400 - \$1900. We are currently working with an experienced industry manufacturer for scale-up and mass production. This collaboration will likely bring down the cost significantly.

Operation cost in the laboratory:

- Cost of PBS buffer (~20 ml is consumed per sample run) = ~US\$ 0.08 per sample
- Cost of calibrant (~3 ml is consumed per sample run) = ~ US\$ 0.18 per sample
- Cost of decontaminant (HOCl; ~20 ml is consumed per sample run) = ~ US\$ 0.04 per sample
- The cost of the MIE biosensor = US\$ 0.1 per sample
- Total operation cost in laboratory = US\$ 0.40 per sample

The MIE biosensor used in the study can be screen printed using a laboratory scale screen printer or industrial screen printers. Therefore, the final cost of the biosensor would depend on the production scale. Moreover, the nanobody used in this electrode has a moderate affinity to the SARS-CoV-2 spike (S) protein to allow for dissociation. This ensures that the nanobody binding capacity of the MIE remains open to accept new SARS-CoV-2 spike (S) protein and thereby prolong the longevity of the MIE biosensor. Based on lab characterization, each MIE can be used for ~70 sample scans. This reusability of the biosensor helps us reduce the cost per sample.

Scaling up the production will significantly reduce the operation and fixed cost of the pAQ monitor. For example, operating a single pAQ monitor in longitudinal virus monitoring mode for a 30-day period and 12 air samples analyzed per hour will work out to ~8700 samples analyzed per month. Such longitudinal virus sampling using a single pAQ will require ~250 electrodes, ~174 L of buffer, and ~26 L of calibrant per month. HOCl consumption is variable and will be based on the user-defined decontamination frequency (~ 3 L a month). If more than 100,000 such units run in parallel at different locations, the monthly demand for the consumable (electrode, calibrant, and buffer) will also be high. If the electrodes are manufactured in bulk using industry-scale screen printers (> 150,000 electrodes a day),⁹ the cost of the MIE biosensor will go down to

<US\$ 0.006 per sample analyzed. Furthermore, the MIE biosensor design can be optimized to help improve the electrode working time. We anticipate the operation life of a single electrode will increase from 70 scans to over 100 scans per electrode after optimization, bringing down the operation cost even lower. Similarly, the cost of the buffer, calibrant, and decontaminant will also go down by 10 folds when purchased in bulk. Overall, based on the production scale, the cost per sample analyzed can be a few cents or even lower.”

A direct cost comparison between the per-sample analysis cost based on the pAQ monitor and RT-qPCR is challenging because this cost will vary greatly based on geographical location and local labor and material cost. Moreover, RT-qPCR requires the patient to either ship their samples to the testing facility or drop the samples in person at the testing facility, which is an extra expense for the patient.

Assuming a typical RT-qPCR sample analysis costs ~US\$100 per nasal swab sample (cost in St. Louis, Missouri, USA), we can see that the operation cost of the lab-built single pAQ monitor per sample is significantly lower (~US\$0.40 per sample). However, if we include the cost associated with collecting environmental air samples and shipping them to the testing facility, this cost will go up even higher. Accurate environmental virus aerosol sample collection is currently an expensive step that requires sophisticated devices and trained personnel to operate the device. Furthermore, in conventional, offline sampling, once the virus aerosol samples are collected, they must be safely stored in a medical-grade refrigerator and then transported to the testing facility in a temperature-controlled container, which increases the overall cost. In contrast, the pAQ monitor performs both the sample collection and virus detection onsite with no sample storage requirements and costs only a fraction of the typical Rt-qPCR cost.

References

1. Darling, T. L. *et al.* mRNA-1273 and Ad26.COV2.S vaccines protect against the B.1.621 variant of SARS-CoV-2. *Med* **3**, 309-324.e6 (2022).
2. Raynor, P. C. *et al.* Comparison of samplers collecting airborne influenza viruses: 1. Primarily impingers and cyclones. *PLoS One* **16**, e0244977 (2021).
3. Luhung, I. *et al.* Experimental parameters defining ultra-low biomass bioaerosol analysis. *npj Biofilms Microbiomes* **7**, 1–11 (2021).
4. Vejerano, E. P. & Marr, L. C. Physico-chemical characteristics of evaporating respiratory fluid droplets. *J. R. Soc. Interface* **15**, (2018).

REVIEWERS' COMMENTS

Reviewer #1 (Remarks to the Author):

Thank you for the in depth correction of the paper and the increase of understanding this revisions provides
The paper can be accepted

Reviewer #2 (Remarks to the Author):

Many thanks for addressing the comments raised. My only other comment would be to ensure the results presented on page 10; line 200 are put into some perspective. As stated in the legend to Supl Figure 8 the Ct values associated with samples taken from infected households are indicative of weakly positive samples with very low RNA concentrations that cannot be accurately quantified.

Reviewer #3 (Remarks to the Author):

This reviewer likes to thank the authors for their effort to address all questions raised by the reviewers.

By doing so they have much improved the manuscript and I have no reservation in recommending it for publication.

Author response:

Reviewer #1 (Remarks to the Author):

Thank you for the in depth correction of the paper and the increase of understanding this revisions provides. The paper can be accepted.

Reply. We thank the reviewer for their encouraging comments and recommendation for publication.

Reviewer #2 (Remarks to the Author):

Many thanks for addressing the comments raised. My only other comment would be to ensure the results presented on page 10; line 200 are put into some perspective. As stated in the legend to Supl Figure 8 the Ct values associated with samples taken from infected households are indicative of weakly positive samples with very low RNA concentrations that cannot be accurately quantified.

Reply. We thanks the reviewer for their comments. We have added the following sentence to give more perspective to the results as suggested by the reviewer.

Page 10, lines 196 – 202 in the revised manuscript:

“Note, the high Ct value (32.7–34.9) measured suggests that the samples collected were weakly SARS-CoV-2 positive and had very low RNA concentration (see Supplementary Method 8), suggesting low virus aerosol shedding by both volunteers, who self-reported as being asymptomatic during the sampling period. These results are consistent with other studies that have also reported low but statistically significant presence of SARS-CoV-2 in COVID patient isolation rooms and highlight the importance of controlling the air transmission of the virus.^{5,11”}.

Reviewer #3 (Remarks to the Author):

This reviewer likes to thank the authors for their effort to address all questions raised by the reviewers. By doing so they have much improved the manuscript and I have no reservation in recommending it for publication.

Reply. We thank the reviewer for their feedback and recommendation for publication.